# Lift-the-flap: what, where and when for context reasoning

## Abstract

Context reasoning is critical in a wide variety of applications where current inputs need to be interpreted in the light of previous experience and knowledge. Both spatial and temporal contextual information play a critical role in the domain of visual recognition. Here we investigate spatial constraints (what image features provide contextual information and where they are located), and temporal constraints (when different contextual cues matter) for visual recognition. The task is to reason about the scene context and infer what a target object hidden behind a flap is in a natural image. To tackle this problem, we first describe an online human psychophysics experiment recording active sampling via mouse clicks in lift-the-flap games and identify clicking patterns and features which are diagnostic for high contextual reasoning accuracy. As a proof of the usefulness of these clicking patterns and visual features, we extend a state-of-the-art recurrent model capable of attending to salient context regions, dynamically integrating useful information, making inferences, and predicting class label for the target object over multiple clicks. The proposed model achieves human-level contextual reasoning accuracy, shares human-like sampling behavior and learns interpretable features for contextual reasoning.

## 1 Introduction

The tiny object on the table is probably a spoon, not an elephant. Objects do not appear in isolation. Instead, they co-vary with other objects, their sizes and colors usually respect regularities with respect to nearby elements, and objects tend to appear at specific locations within a scene. Humans exploit these contextual associations. Contextual analyses based on the statistical summary of object relationships, provide an effective source of information for perceptual inference tasks, such as object detection (Torralba (2003); Park et al. (2010); Hoiem et al. (2005); Torralba et al. (2010); Liu et al. (2018b)), scene classification (Gonfaus et al. (2010); Torralba et al. (2005); Yao et al. (2012)), semantic segmentation (Yao et al. (2012)), and visual question answering (Teney et al. (2017)).

An example of how contextual information is incorporated during object recognition is lift-the-flap books, where a flap covers part of the page. Children make guesses about what is behind the flap based on the context and check their answers by lifting the flap (Figure 1a). Here we investigate *what* image features matter for contextual reasoning and *where* those features are with respect to the target object of interest. Furthermore, scene interpretation in humans involves a sequence of eye movements (Zhang et al. (2018)), each one of these image samples providing additional context to inform interpretation of the contents of the next location. Therefore, we also investigate whether integration of scene information over time (*"when"* information) matters for context reasoning.

To tackle the problem of contextual reasoning, we introduce the lift-the-flap task and conduct online psychophysics experiments where subjects make mouse clicks while they explore important contextual cues to identify a hidden target (Figure 1b). We investigate the contextual reasoning strategies observed from human active clicking patterns. As a proof of concept, we propose a recurrent attention model (ClickNet), to automatically learn these contextual reasoning strategies. The model guides attention towards regions with informative context, decides where to click on the image, and makes inferences about the target behind the flap. The learnt clicking patterns and predicted class labels by ClickNet share remarkable similarities with human behavior.

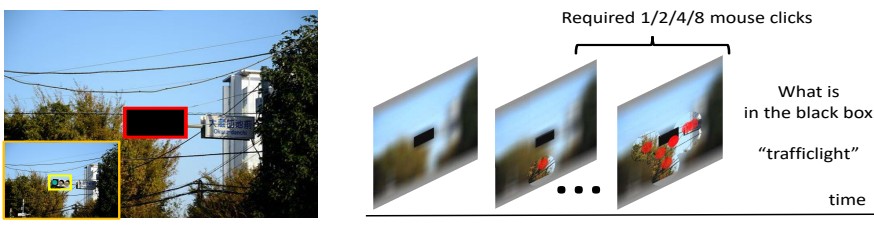

(a) Lift-the-flap game          (b) Schematics of Behavioral Experiment

Figure 1: **Schematic of the lift-the-flap task and human behavioral experiment**. (a) The task requires subjects to capitalize on the scene context in a natural image to infer what is behind the black box (the hidden target). The original image (bottom left) reveals the target object ("traffic light"); this image was *not* shown in the actual experiments. (b) A blurred image with the hidden target was presented to the subject. To identify contextual areas of importance surrounding the hidden target, subjects used the computer mouse to click on image a pre-specified number of times (red dots). Upon clicking a certain location, a circle of fixed radius was revealed at high resolution. After the required number of clicks, subjects typed a noun describing the hidden target object. The experiments were conducted online using Amazon Mechanical Turk Turk (2012) on 100 subjects, 50 trials per subject. Figure 3 also shows results for a variation of the experiment conducted in the lab while tracking eye movements.

## 2 RELATED WORKS

### 2.1 ROLE OF CONTEXT IN HUMAN VISION

Contextual information affects the efficiency of several visual processes (Auckland et al., 2007; Biederman et al., 1982; Hollingworth, 1998; Bar & Aminoff, 2003; Goh et al., 2004; Aminoff et al., 2006), such as object recognition (Auckland et al., 2007), object detection (Biederman et al., 1982; Hollingworth, 1998), visual working memory (Friedman, 1979; Aminoff et al., 2006) and visual search (Henderson et al., 1999). Objects appearing in a familiar background can be detected more accurately and processed more quickly than objects appearing in an incongruent scene. Here we focus on what visual features contribute to contextual reasoning, which parts of image regions attract human attention to make inferences, and the dynamic sequence of directed sampling needed to idenfity a hidden target.

### 2.2 ROLE OF CONTEXT IN COMPUTER VISION

Contextual reasoning about objects and relations is critical to machine vision. In fact, many object recognition studies using natural image datasets such as ImageNet, rely implicitly but strongly on contextual feature regularities Geirhos et al. (2018); Brendel & Bethge (2019). Some works also show that these models can fail when objects are placed in an incongruent context (Beery et al. (2018); Dvornik et al. (2018); Choi et al. (2012)). These studies motivate examining the role of pure context *without any object information* in situations of complete occlusion (lift-the-flap); here we provide human and computational benchmarks for object inference using exclusively context information. Several studies use context to improve object detection (Park et al., 2010; Hoiem et al., 2005; Torralba et al., 2010; Liu et al., 2018b; Chen et al., 2018b). Context can take the form of global scene context (Torralba et al., 2010), ground plane estimation (Park et al., 2010), geometric context (Hoiem et al., 2005), relative location Desai et al. (2011), 3D layout (Lin et al., 2013), and spatial support and geographic information (Divvala et al., 2009). Researchers proposed Conditional Random Field (CRF) models that reason jointly across multiple computer vision tasks in image labeling and scene classification Gonfaus et al. (2010); Yao et al. (2012); Ladicky et al. (2010). Context can lead to improved performance in both object detection and semantic segmentation tasks Mottaghi et al. (2014). Neural network architectures incorporating contextual information have been used in object priming (Torralba, 2003), place and object recognition (Wu et al., 2018; Torralba et al., 2005), object detection (Liu et al., 2018b; Chen et al., 2018b), and visual question answering (Teney et al., 2017). Here we focus on developing a biologically inspired computational model for contextual reasoning that can automatically and dynamically sample image regions of interest, integrating information in space and time to make inferences about a hidden object. Additionally, we compare the model's performance against human behavior in the same task.

Several interesting approaches have combined graphical models with deep neural networks for structural inference, primarily in structured prediction tasks (Marino et al., 2016; Choi & Savarese, 2012; Chen et al., 2018a; Teney et al., 2017; Hu et al., 2016; Battaglia et al., 2016; Xu et al., 2017; Chen et al., 2018b; Choi et al., 2012). Hu et al. (2016) designed a structured model to improve classification performance by leveraging relations among scenes, objects, and their attributes. A structured inference model is also used in (Choi & Savarese, 2012; Deng et al., 2016) to analyze relations in group activity recognition. Several works, like Structural-RNN Jain et al. (2016) and Interaction Net Battaglia et al. (2016), combine spatiotemporal graphs and sequence learning. These works assume full contextual information is available, while in our experiment we consider only partial contextual information that is sequentially revealed after each mouse click. DeepLab sends the response at the final layer of a CNN to a CRF model for semantic image segmentation Chen et al. (2018a). Subsequently, Schwing & Urtasun (2015); Zheng et al. (2015) transformed the CRF model into a Recurrent Neural Network. Breaking away from these studies where graph optimization is performed globally, our proposed model selects important visual features using an attention mechanism and integrates partial information over multiple steps, which is computationally more efficient and accurate in the current task (Section 5).

## 3  LIFT-THE-FLAP TASK

### 3.1  HUMAN BEHAVIORAL EXPERIMENTS

Subjects were presented with a natural image where one object was hidden by a rectangular black box and everything else was blurred. They were allowed a fixed number of mouse clicks between 1 and 8, each click revealed part of the image in high resolution. To minimize overlap between clicks, we enforced subjects to click at places at least 10 pixels apart from all previous clicks. After the target number of clicks, they had to provide a single word to describe the object hidden behind the black box (Figure 1b). The clicking experiments were run on Amazon Mechanical Turk Turk (2012). The stimulus set consisted of 573 images from the test set of the MSCOCO Dataset (Lin et al., 2014), spanning 80 object categories. This dataset has been widely used for object recognition and detection studies (Lin et al., 2014). We constrained the stimulus set to have a uniform distribution of 6 - 8 target objects per category. To avoid any potential memory effects, subjects were only exposed to each image once. The trial presentation order was randomized.

### 3.2  GROUND TRUTH RESPONSES

In contrast to experiments where subjects are forced to perform N-way categorization (e.g., Tang et al. (2018)), here there were no constraints on how subjects describe the hidden object. This probing mechanism was implemented for two reasons: (i) it is difficult for humans to memorize 80 object classes and there could be non-uniform memory effects impacting the results; (ii) presenting humans with an 80-choice question could introduce biases in their inference processes.

We could not simply use the 80 category labels to evaluate performance because subjects could use other words or synonyms and we are interested in context reasoning rather than language abilities. Therefore, to evaluate humans' performance, we separately collected a distribution of ground truth answers for each hidden target by presenting to 10 *other* subjects, who did not participate in the main task, the same set of images with the target objects highlighted by a bounding box but not hidden. During the lift-the-flap task, a response was considered to be correct if it matched any of the ground truth labels, allowing for plurals and misspellings.

### 3.3  EVALUATION METRICS

We introduce several evaluation metrics to measure contextual reasoning accuracy and to compare the consistency of mouse clicking patterns between humans and computational models. We evaluated ClickNet on the MSCOCO Dataset using the typical classification accuracy measure. In Fig 5b, we report **top-1 classification accuracy** as a function of **context-object ratio**. The context-object ratio is defined as the total area of the image *excluding* the hidden target divided by the hidden target object size. For example, a context-object ratio of 1 implies that the size of the black box and the size of the contextual information is the same (see Figure 5b for example images with different context-object ratios).

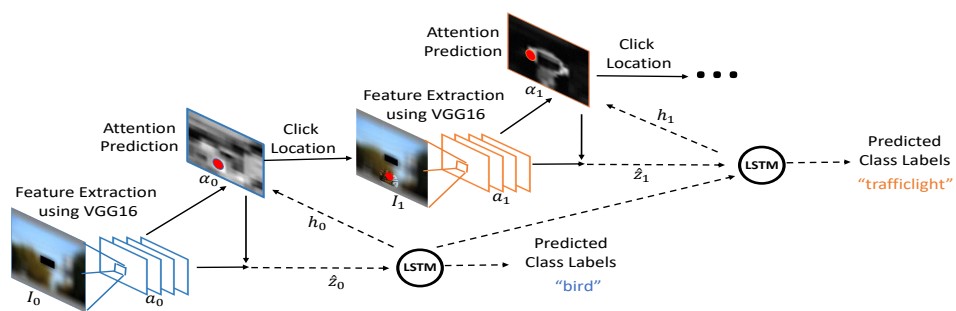

Figure 2: **Architecture overview of the ClickNet model**. The diagram depicts the iterative modular steps carried out by ClickNet for contextual reasoning over multiple clicks in the lift-the-flap task (Figure 1b). ClickNet consists of 3 main modules: feature extraction, attention, and recurrent memory. For illustrative purposes, only the first and second clicks in a trial are shown here. ClickNet performs feature extraction using the VGG16 network pre-trained on ImageNet and produces feature maps $a_0$. Conditioned on the hidden state $h_0$ and feature maps $a_0$, ClickNet produces an attention map $\alpha_0$, which is used to select the next click location (red dots) and to modulate the feature maps for contextual reasoning (Figure S1). The recurrent network in the LSTM module (Figure S2) integrates over time the attentionally modulated feature maps $\widehat{\mathbf{z}_0}$, and outputs a predicted class label after each click (here "bird" and "trafficlight" before the 1st and 2nd clicks). After the first click, the input image gets updated with parts at clicked locations revealed in high resolution. These three modular steps repeat until the specified number of clicks have been made. See supplementary figures S1 and S2 for implementation details on the attention and LSTM modules, respectively.

To measure the degree of spatiotemporal consistency between two mouse clicking patterns (human versus human; or human versus computational models), we introduced two measures. First, we computed the minimum Euclidean distance between the sequence of clicks in each trial, regardless of order. The smaller the median in the distribution of distances, the more similar the two mouse clicking patterns are. Second, we introduced a click sequence score, as originally defined in evaluating eye fixation sequence consistency Madsen et al. (2012); Zhang et al. (2018). Here, we use the same metrics to compare the spatial-temporal consistency between click sequences. The larger the click sequence score, the more similar the sequences are.

## 4 CLICKNET ARCHITECTURE

We propose a recurrent neural network for context reasoning (ClickNet), extending previous work on image captioning (Xu et al., 2015). ClickNet integrates attention-modulated context information over multiple clicks, makes a decision about the next click location based on the attention map, and infers the class label of the hidden target after every click (Figure 2).

As in the human psychophysics experiment (Figure 1b), ClickNet is first presented with a blurred image $I_0$, which is the original image $\mathbf{I}$ with uniform gaussian blur and where the target object is covered by a black box. ClickNet makes the first attempt to predict a class label $y_0$ out of a pre-defined set of $C$ object classes and decides its first click location $m_1$. In every trial, over a series of $T$ clicks, the input image $I_t$ to ClickNet gets updated with circular regions of constant radius $R$ centered at all previous click locations $M = \{m_1, ...m_T\}$, revealed in its original resolution in $\mathbf{I}$. The black box is constant and none of its content is ever revealed, even if the model opts to click within the box or if the circle centered on the click encompasses part of the box.

### 4.1 CONVOLUTIONAL FEATURE EXTRACTION

At each time $t$ where $t \in \{0, ..., T\}$, ClickNet takes $I_t$ as input and uses a feed-forward convolutional neural network to extract feature maps $a_t$. We use the VGG16 network (Simonyan & Zisserman, 2014), pre-trained on ImageNet (Deng et al., 2009). Consistent with previous works showing that these feed-forward convolutional neural networks for visual recognition tasks are vulnerable to domain shift such as image blur and noise when those transformations do not exist in the images used for training (Liu et al. (2018a); Dodge & Karam (2016)), we observe that it is necessary to

fine-tune the pre-trained VGG16 network for lift-the-flap task (see implementation details in 4.4 and Section A). To focus on specific parts of the image and select features at those locations, we have to preserve the spatial organization of features; thus, ClickNet uses the output feature maps at the last convolution layer of VGG16.

A feature vector $\mathbf{a_{ti}}$ of dimension $D$ represents the part of the image $I_t$ at location $i$, where $i = 1, .., L$ and $L = W \times H$ and $W$ and $H$ are the width and height, respectively, of the feature map:

$$a_t = \{\mathbf{a_{t1}}, ..., \mathbf{a_{tL}}\}, \quad \mathbf{a_{ti}} \in \mathbb{R}^D \tag{1}$$

## 4.2 ATTENTIONAL MODULATION

We use a "soft-attention" mechanism as introduced by Ba et al. (2014) to compute "the context gist" $\widehat{\mathbf{z}_t}$ on $I_t$ (Figure S1). For each location $i$ in $a_t$, the attention mechanism generates a positive scalar $\alpha_{ti}$, representing the relative importance of the feature vector $\mathbf{a_{ti}}$ for context reasoning. This relative importance $\alpha_{ti}$ depends on the feature vectors $\mathbf{a_{ti}}$, combined with the hidden state at the previous step $\mathbf{h_{t-1}}$ of a recurrent network described below.

$$e_{ti} = A_h \mathbf{h_{t-1}} + A_a \mathbf{a_{ti}}, \quad \alpha_{ti} = \frac{\exp(e_{ti})}{\sum_{i=1}^L \exp(e_{ti})} \tag{2}$$

where $A_h \in \mathbb{R}^{1 \times n}$ and $A_a \in \mathbb{R}^{1 \times D}$ are weight matrices initialized randomly and to be learnt. Instead of addition, an alternative is to use element-wise multiplication: $A_h \mathbf{h_{t-1}} \circ A_a \mathbf{a_{ti}}$. Empirically, we did not observe any performance difference with such a multiplicative term (see Section A). Because not all attended regions might be useful for context reasoning, the soft attention module also predicts a gating vector $\beta_t$ from the previous hidden state $h_{t-1}$, such that $\beta_t$ determines how much the current observation contributes to the context vector at each location: $\beta_t = \sigma(W_\beta \mathbf{h_{t-1}})$, where $W_\beta \in \mathbb{R}^{L \times n}$ is a weight matrix and each element $\beta_{ti}$ in $\beta_t$ is a gating scalar at location $i$. As also noted by Xu et al. (2015), $\beta_t$ helps put more emphasis on the salient objects in the images. Once the attention map $\alpha_t$ and the gating scale $\beta_t$ are computed, the model applies the "soft-attention" mechanism to compute $\widehat{\mathbf{z}_t}$ by summing over all the $L$ regions in the image:

$$\widehat{\mathbf{z}_t} = \sum_{i=1}^L \beta_{ti} \alpha_{ti} \mathbf{a_{ti}} \tag{3}$$

The next click location $m_{t+1}$ corresponded to the maximum on the attention map:

$$m_{t+1} = \arg\max_i \alpha_{ti} \tag{4}$$

The attentional module is smooth and differentiable and ClickNet can learn all the weight matrices in an end-to-end fashion via back-propagation.

## 4.3 RECURRENT CONNECTIONS USING LONG SHORT-TERM MEMORY (LSTM)

We use a long short-term memory (LSTM) network to output a predicted class label $y_t$ based on the previous hidden state $\mathbf{h_{t-1}}$ and the context gist vector $\widehat{\mathbf{z}_t}$ for $I_t$ (Figure S2). Our implementation of LSTM closely follows Zaremba et al. (2014) where $T_{s,t} : \mathbb{R}^s \to \mathbb{R}^t$ defines a linear transformation with learnable parameters. The variables $\mathbf{i_t}, \mathbf{f_t}, \mathbf{c_t}, \mathbf{o_t}, \mathbf{h_t}$ represent the input, forget, memory, output and hidden state of the LSTM respectively:

$$\begin{pmatrix} \mathbf{i_t} \\ \mathbf{f_t} \\ \mathbf{o_t} \\ \mathbf{g_t} \end{pmatrix} = \begin{pmatrix} \sigma \\ \sigma \\ \sigma \\ \tanh \end{pmatrix} T_{D+n,n} (\widehat{\mathbf{z}_t}, \mathbf{h_{t-1}}) \tag{5}$$

$$\mathbf{c_t} = \mathbf{f_t} \odot \mathbf{c_{t-1}} + \mathbf{i_t} \odot \mathbf{g_t}, \quad \mathbf{h_t} = \mathbf{o_t} \odot \tanh(\mathbf{c_t}) \tag{6}$$

where $n$ is the dimensionality of LSTM, $\sigma$ is the logistic sigmoid activation, and $\odot$ indicates element-wise multiplication.

To cue ClickNet about the location of the hidden target, we initialize the memory state $\mathbf{c_0}$ and hidden state $\mathbf{h_0}$ of the LSTM based on a binary mask that contains zeros everywhere and ones in the hidden target location. Specifically, $\mathbf{c_0}$ and $\mathbf{h_0}$ are predicted by an average of all feature vectors $a_0$ over all $L$ locations with two separate linear transformations $W_{c0} \in R^{n \times D}$ and $W_{h0} \in R^{n \times D}$:

$$\mathbf{c_0} = W_{c0}(\frac{1}{L} \sum_i^L \mathbf{a_{0i}}), \quad \mathbf{h_0} = W_{h0}(\frac{1}{L} \sum_i^L \mathbf{a_{0i}}) \tag{7}$$

To predict the class label $y_t$ of the hidden target, the LSTM computes a classification vector where each entry denotes a class probability given the hidden state:

$$y_t = \arg \max_c p(y_c), \quad p(y_c) \propto L_h \mathbf{h_t} \tag{8}$$

where $L_h \in \mathbb{R}^{C \times n}$ is a matrix of learnt parameters initialized randomly.

### 4.4 TRAINING AND IMPLEMENTATION DETAILS

We trained ClickNet end-to-end by minimizing the cross entropy loss between the predicted label $y_t$ at each time step $t$ and the ground truth label $x$. In contrast with previous work where there is a regularization term $\sum_t \alpha_{ti} = 1$ in the loss function to encourage the model to acquire as much information all over the image by exploration (Xu et al. (2015)), we did not find a significant performance increase in the lift-the-flap task when adding such a term (see Section A). Hence, let ClickNet freely explore the image without any such constraints. Since we do not impose any constraints on the next click locations, ClickNet might make decisions to click at the previously clicked locations.. The loss function is definef by:

$$LOSS = \sum_{t=1}^T (-\log(P(y_t|x))) \tag{9}$$

We used all images from the MSCOCO training set for training and validating all models. On every training image, each object can be blocked out as the hidden target. The input image size to ClickNet was $400 \times 400$ pixels. We used a Gaussian filter of size $51 \times 51$ with variance $64$ pixels to blur the images. The radius $R$ of the circular region revealed by each click was $55$ pixels. As in the human psychophysics experiments (Fig 1b), in each trial, we set the total number of time steps $T = 8$ for training ClickNet (ClickNet predicts the label after the 1st click at $T = 1$). The dimension of the LSTM module was $n = 512$. The feature maps extracted from the last convolution layer was of size $2048 \times 28 \times 28$, and the total number of locations was $L = 28 \times 28 = 784$. The Adam optimizer (Kingma & Ba, 2014) was used with a learning rate of $10^{-4}$ to fine-tune the VGG16 network, and a learning rate of $4 \times 10^{-4}$ to train the attentional module and the LSTM module. The network was developed in Pytorch, based on (Xu et al., 2015). All source code for our proposed architecture, and the data from the psychophysics experiments will be released publicly upon publication.

### 4.5 VARIATIONS OF PROPOSED NETWORK ARCHITECTURE AND COMPARATIVE METHODS

Previous work has shown that it is possible to augment vision systems with human perceptual supervision on many difficult computer vision tasks, such as Vondrick et al. (2015); Kovashka et al. (2016). One central goal in our study is to investigate what, where, and when matter for human contextual reasoning in the lift-the-flap game and whether these factors could help improve current machine learning algorithms. We now introduce two variations of ClickNet with human inputs at the **testing** stage:

**ClickNet-humanclick.** Instead of clicking at the location with highest activation value on the attention map predicted by ClickNet, we substitute the input with human clicking images.

**ClickNet-RandPrior.** We observe strong spatial bias in the human clicking patterns where most of the clicks tend to be nearby the hidden target (see Sec 5.4). To test whether this is sufficient to account for human behavior, we generated random clicks surrounding the black bounding box and used the resulting images with a strong spatial clicking prior as inputs to ClickNet. We set the same criterion to consider two clicks as overlapping as in the human behavioral experiments; the probability that two clicks overlap in this model is 0.0065.

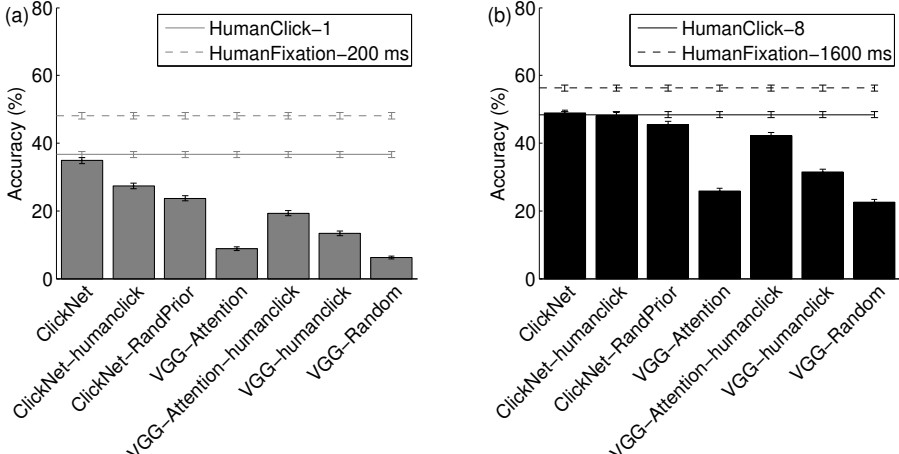

Figure 3: **Contextual reasoning accuracy of humans and models**. Performance for humans (horizontal lines) and models (bars) for **(a)** 1 click (gray), and **(b)** 8 clicks (black) (Fig S4 shows results for 2 and 4 clicks, and additional comparison models). Section 3.3 defines the evaluation metric and Section 4.5 describe each model. Error bars denote SEM across images.

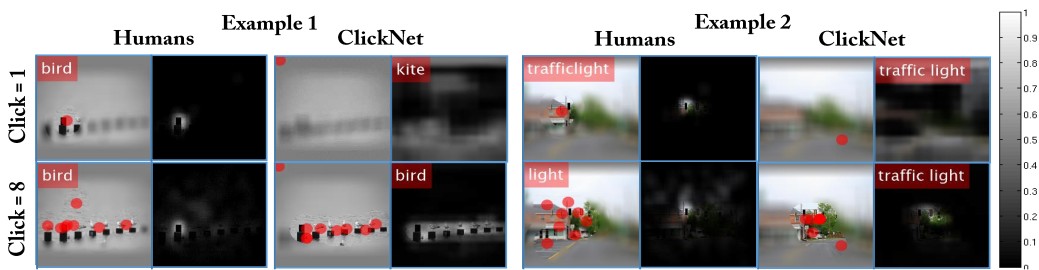

Figure 4: **Example visualization for humans and ClickNet**. Two example trials (first four columns is example 1, last four columns is example 2), either with 1 click (rows 1) or 8 clicks (rows 2), for one human (columns 1, 2, 5, 6) or ClickNet (column 3, 4, 7, 8) (Fig S3 shows results for 2 and 4 clicks). Red dots denote clicked locations. Top-left corner shows output labels after the required number of clicks. Column 2 and 6 show the mouse click maps aggregated over subjects. Brighter regions denote more mouse clicks (see scale bar on right). Column 4 and 8 show the attentional map predicted by ClickNet. Brighter regions denote higher attentional values.

To gauge how much contextual information helps object recognition, it is interesting to isolate the effect of context and object, analyze them separately and then study both combined. The lift-the-flap task provides a benchmark to study the contextual reasoning problem alone. In addition, we evaluate ClickNet when both the object region and the context are revealed to ClickNet (an upper bound) and when only the object region (the tightest bounding box encompassing the object) is revealed to ClickNet (Section D).

To study the role of attention and recurrent connections, we introduced two ablated models.

**Variations of VGG16.** One intuitive way of solving the context reasoning problem is to use a feed-forward object recognition network pre-trained on ImageNet, e.g. VGG16 (Simonyan & Zisserman, 2014), and fine-tune it to classify the hidden target on MSCOCO dataset. During training, the input to the network was an image where one object on the image was randomly covered with a black bounding box. We tested the performance of this alternative model on the 573 images selected for human psychophysics experiments with different input variations: human clicking images (**VGG-humanclick**), the blurred images (**VGG-Blur**), the full-resolution images (**VGG-Fullres**) and images with random clicks (**VGG-Random**).

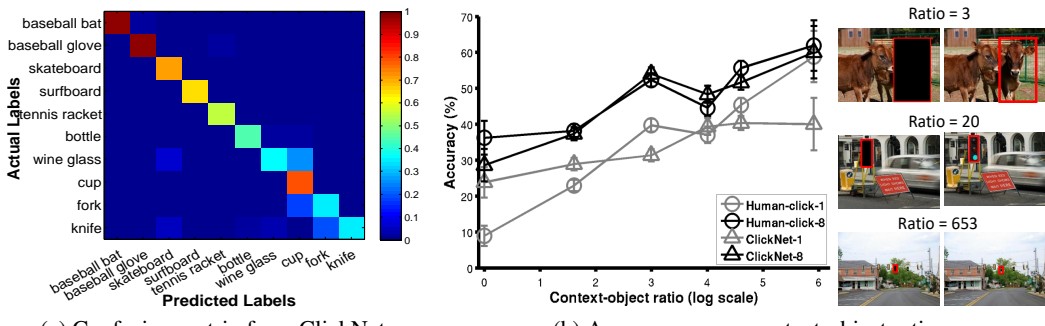

| (a) Confusion matrix from ClickNet | (b) Accuracy versus context-object ratio |

Figure 5: **Improvement in contextual reasoning accuracy with context-object ratio and patterns of mistakes**. (**a**) Partial view for illustrative purposes of confusion matrix showing the mistakes made by ClickNet among 10 of the 80 object categories in MSCOCO (Fig S5 shows the complete confusion matrix with all 80 categories). The element in row $i$, column $j$ denotes the probability that ClickNet predicted label $j$ while the ground truth label was $i$ (see scale bar at right). The sum of probabilities in a row in the full confusion matrix (but not here) equals 1. (**b**) Human (circle) and model (triangle) accuracy for 1 click (gray) and 8 clicks (black) as a function of context-object ratio, shown in logarithmic scale (Sec. 3.3). Right: 3 example images with different context-object ratios. Only the images on the first column were shown (the second column is shown here for illustrative purposes only). Error bars indicate SEM across images.

**VGG-Attention.** Previous work has demonstrated the efficiency of attention in computer vision tasks (Nguyen et al., 2018), such as question answering and image captioning Xu et al. (2015). To study the effect of attention in contextual reasoning, we added an attention module to the end of VGG16. To make the complexity of the architecture comparable with ClickNet, we added the same number of fully connected layers as in the LSTM module. As in ClickNet, we used the location with the highest activation value on the attention map to predict the next click. VGG-Attention takes the updated image as input and iteratively predicts the hidden target label. In contrast to ClickNet, the network is feed-forward and there is no incorporation of past information integrated over clicks. We also test VGG-Attention with human clicks (**VGG-Attention-humanclick**) and randomly generated click locations with strong spatial priors (**VGG-Attention-RandPrior**).

We also considered several competitive baselines and existing methods of modeling temporal dynamics. These include **Human-fixations**, **SVM-category**, **SVM-category-instances**, **Hidden Markov Model (HMM)** and **DeepLab-Conditional Random Field (CRF)**. See Section B for these comparison methods.

## 5 RESULTS

### 5.1 WHAT: REGION SELECTION VIA ATTENTION AND PRIOR INFORMATION

Subjects inferred the identity of the hidden target object with $36.7 \pm 0.9\%$ accuracy after 1 click (Fig 3, gray horizontal line). Performance showed a small, but significant improvement when allowing subjects 8 clicks, reaching $48.4 \pm 0.9\%$ (Fig 3, black horizontal line, $p < 10^{-9}$, two-tailed t-test, t=-6, df=2593). In-lab experiments corroborated these results showing accuracies of $48.1 \pm 0.9\%$ after 200 ms exposure and $56.3 \pm 0.9\%$ after 1,600 ms exposure to the images.

The same images that subjects saw were used to evaluate ClickNet (Figure 3). The ClickNet model showed a close approximation to human performance in the mturk experiments, reaching a top-1 classification performance of $34.9 \pm 0.9\%$ for 1 click and $48.87 \pm 0.9\%$ for 8 clicks. In both cases, performance was only slightly lower than human performance in 1 click ($p = 0.47$, two-tailed t-test, t=0.73, df=1828) and 8 clicks ($p = 0.86$, two-tailed t-test, t=0.17, df=1909). Performance for intermediate numbers of clicks is shown in (Figure S4). For all the computational models, random guessing would yield accuracy = $1.25\%$.

The worst performing model, VGG-Random, yielded performance above chance levels, emphasizing that even small amounts of high-resolution contextual data at arbitrary locations can help solve the problem. Yet, VGG-Random was well below ClickNet's performance ($p < 10^{-15}$, two-tailed t-test, t=21, df=3028). Adding attention to the model (VGG-Attention) yielded only

minimal improvement. An important ingredient missing in VGG-Random and VGG-Attention is the informed location of the clicks. Humans and ClickNet do not sample the image randomly, but rather explore informative locations. Figure 4 shows qualitative examples of clicking patterns from humans and ClickNet. Both humans and ClickNet attend to salient regions on the images. For example, clicks often occur near birds in the first example and near traffic lights in the second example. Accordingly, substituting the random clicks for the human clicks into the VGG models (VGG-humanclick and VGG-Attention-humanclick) yielded a large performance boost. Conversely, substituting the ClickNet clicks with random clicks leads to large drop in performance when there is only 1 click, even when we artificially try to boost performance by constraining the clicks to be near the hidden target object (ClickNet-RandPrior). This effect is also evident with 2 clicks and 4 clicks (Fig. S4), but disappears with 8 clicks because there is already a lot of high resolution information in the image surrounding the hidden target, and ClickNet can integrate information over time to capitalize on it. Interestingly, the ClickNet sampling clicks are sufficiently close to human clicks that substituting the ClickNet clicks with human clicks does not improve performance (ClickNet-humanclick).

We considered several other comparative models (Fig. S4). Interestingly, using just a few clicks, ClickNet reaches performance that is essentially equivalent to that of VGG using a full resolution version of the entire image. Other comparative models (VGG-Blur, SVM-category, SVM-category-instances, HMM, DeepLab-CRF) showed above chance performance but their accuracies were well below ClickNet.

## 5.2 WHAT: THE MORE, THE MERRIER

To investigate how much context information is needed to enhance recognition, we evaluated accuracy as a function of context-object ratio (Fig 5b, Sec 3.3). Images with higher context-object ratio contained more context information for inference, and yielded higher accuracy both for humans and models. Similarly, accuracy improved with increasing numbers of clicks (Fig 3 and Fig 5b).

It is not just the quantity of context, but also the specific quality of contextual information that matters. In the real world, objects do not tend to appear in isolation but rather they co-vary with other objects. As ClickNet explores more regions on the image, it integrates information at previous clicked locations and learns associations of objects. The pattern of mistakes made by ClickNet is indicative of those associations (Fig 5a and Fig S5). ClickNet often makes "reasonable" wrong guesses when there is ambiguity in context reasoning, as humans do. For example, knife tends to be associated and therefore confused with spoon, fork, and wine glass, but knife seldom co-occurs with baseball bat or skateboard in these images.

## 5.3 WHERE: CONSISTENCY OF HUMAN AND MODEL CLICKS

We hinted at the similarity in the clicking patterns between humans and ClickNet based on the accuracy of the ClickNet-humanclick and ClickNet-RandPrior models in Fig 3. To more directly assess whether ClickNet learned to sample the image to gather information about areas of contextual relevance, we directly quantified the similarity in clicking patterns (Figure 6 and S6). To interpret the distance between human and model clicks, we computed the degree of human-human consistency in the clicking patterns. The clicking patterns of ClickNet were overall similar to those made by humans. The model clicks were still different from the consistency between two humans (8 clicks: $p < 10^{-15}$, two-tailed t-test, t=36, df=29542); yet, the model clicks were more similar to human clicks than random clicks (8 clicks: $p < 10^{-15}$, two-tailed t-test, t=44, df=35310).

## 5.4 WHERE: TENDENCY OF CLICKING NEARBY THE TARGET

There was a strong spatial bias towards clicking near the target for both humans and ClickNet (e.g., Figure 4). To quantify this spatial bias, we computed the Euclidean distance between the clicked locations and the center of the bounding box, normalized by the diagonal of the bounding box (Figure 6c). Humans tended to click within one diagonal distance of the target box. Interestingly, although ClickNet does not take any human supervision during training, ClickNet still learned to capture the tendency of clicking near the target. We asked whether this spatial bias in sampling behavior is sufficient to explain performance in this task in a modified version of ClickNet. We removed the clicks dictated by the attention module and instead forced the clicks to be randomly distributed while still respecting the spatial distribution in Fig 6c (ClickNet-RandPrior). Both ClickNet and ClickNet-humanclick surpassed ClickNet-RandPrior by 28.3% and 7% over all click conditions respectively (Figure 3). Similar results were obtained when using only the VGG

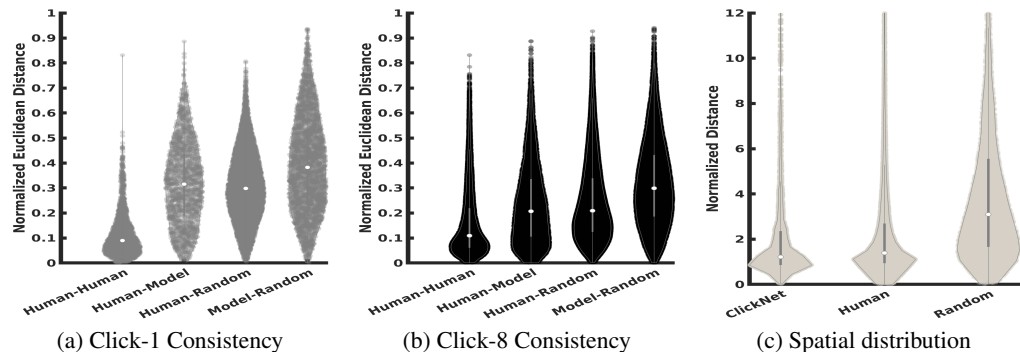

(a) Click-1 Consistency        (b) Click-8 Consistency        (c) Spatial distribution

Figure 6: **Model click locations were similar to human sampling.** (**a-b**) Consistency of click patterns between human subjects (human-human), consistency between humans and model (human-model), consistency between humans and random (human-random) and consistency between models and random (model-random) for 1 click (**a**) and 8 clicks (**b**) measured by the distribution of normalized Euclidean distances with respect to the diagonal of the image between any pairs of clicks by humans and ClickNet or random clicks. In each trial, we permute the sequence of mouse clicks between pairs of human and model clicks such that their sum of Euclidean distance is minimized across all clicks. The white circles denote the median of the distribution and the light grey bar denote the 1st and 3rd quartiles. (**b**) Euclidean Distance between click locations and center of the hidden target bounding box normalized by the diagonal of the hidden target bounding box.

architecture: VGG-Attention-humanclick was 16% better than VGG-Attention-RandPrior (Fig S4). Therefore, the spatial bias in clicking behavior cannot account for performance: sampling for context reasoning involves more than clicking near the target.

## 5.5 WHEN: TEMPORAL INTEGRATION HELPS RECOGNITION

Several lines of evidence support the importance of the recurrent network in the LSTM module in ClickNet. ClickNet outperformed the competitive baselines and state-of-the-art comparative methods to make inferences (Figure 3 and Fig S4). We tested whether the co-occurrence of object categories or the number of objects per category present in the image would be sufficient for context reasoning (SVM-category and SVM-category-instances). Even though these alternative models were exposed to full contextual information on the image and assumed perfect labeling of all objects in the image (except for the hidden target object), there was still a large overall performance drop in performance with respect to ClickNet. Moreover, graphical models for inference, such as Hidden Markov Model and DeepLab with Conditional Random Field (Sec 4.5) failed to reach ClickNet's accuracy in this task (Fig S4). The ablation studies eliminating the LSTM module further support the role of integrating information over multiple clicks in this task, as evidenced by the observation that ClickNet outperformed the VGG-Attention model.

## 6 DISCUSSION

Here we quantitatively studied the role of contextual information in visual recognition in human observers and computational models in a task that involved inferring the identity of a hidden target object. Context influenced recognition based on the amount of context, the specific location of contextual cues, and the dynamic integration of salient visual features. We introduced a recurrent neural network model that combines a feed-forward visual stream module that extracts image features in a dynamic fashion, combined with an attention module to prioritize different image locations and select the next click location, and a recurrent LSTM module that integrates information over time and produces a label for the hidden object. Surprisingly, even though the model lacks the expertise that humans have in interacting with objects in their context, the model approximates human sampling behavior (Click consistency in spatial domains (Euclidean distance distribution and spatial bias in Figure 6) and temporal domains (click sequence similarity score in Figure S6)) and reaches almost human-level performance in this contextual reasoning task (contextual reasoning accuracy in Figure 3 and Figure 5b) and reaches almost human-level performance in this contextual reasoning task. The model opens the doors to examine more complex form of reasoning about scenes and how to integrate sequential sampling with prior knowledge.

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

APPENDIX

We provide supplementary figures and materials here. All labels in supplementary figures and tables are pre-fixed with letter S in front.

## A   INTRODUCTION TO ADDITIONAL ABLATED MODELS AND CONTROLS

**ClickNet-noFineTune.** As explained in Section 4.1, there is a domain shift problem in feed-forward convolutional neural networks for visual recognition tasks when trained using one type of images (e.g., without blurring) and tested using other types of images (blurred). To quantify this effect, here we introduce **ClickNet-noFineTune** where the weights of the feature extractor module were loaded from the VGG16 network pre-trained on ImageNet and were fixed during training. The results are reported in Figure S4. There was a performance drop of around 13% with 1 click and 15.1% with 8 clicks, compared with ClickNet.

**ClickNet-WithAlphaLoss.** Previous work included an additional alpha loss term in equation 9 (Xu et al. (2015)). To study the effect of this alpha loss term as explained in Section 4.4, we introduced **ClickNet-WithAlphaLoss**. The updated loss function becomes:

$$LOSS = \sum_{t=1}^{T}(-\log(P(y_t|x))) + \lambda \sum_{i}^{L}(1 - \sum_{t}^{T} \alpha_{ti})^2 \tag{10}$$

We chose the hyperparameter $\lambda = 2$ in this equation. The other training specifications remain the same. The result is reported in Figure S4. There was a performance drop of around 2% with 1 click and 4% with 8 clicks compared with ClickNet. Results presented throughout the main text do not include this alpha term.

**ClickNet-AttenMulti.** In Section 4.2, we additively combined $\mathbf{h_{t-1}}$ and $\mathbf{a_{ti}}$. An alternative way of computing attention is to combine $\mathbf{h_{t-1}}$ and $\mathbf{a_{ti}}$ via multiplication. As mentioned in Section 4.4, we introduced **ClickNet-AttenMulti** and empirically test this possibility. The attention computation in Equation 2 was updated as follows:

$$e_{ti} = A_h \mathbf{h_{t-1}} + A_a \mathbf{a_{ti}} + A_h \mathbf{h_{t-1}} \circ A_a \mathbf{a_{ti}} \tag{11}$$

where $\circ$ denotes element-wise multiplication. The results are reported in Figure S4. There was a performance drop of around 1% with 1 click and 4% with 8 clicks compared with ClickNet.

## B   INTRODUCTION TO COMPETITIVE BASELINES AND EXISTING METHODS

**Human-fixations.**   We were concerned about the variable viewing conditions in the MTurk experiments. Therefore, we conducted in-lab psychophysics measurements as a benchmark. In the in-lab experiment, after 500 ms fixation, a bounding box with a fixation cross in the center presented for 1,000 ms indicated the target position and cued subjects to attend to the hidden target location. To ensure that in-lab subjects paid attention to the hidden target location, we recorded their eye movements using an EyeLink D1000 system (SR Research, Canada). The image with the black box was shown for 200, 400, 800, or 1600 ms. Subjects freely moved their eyes; after stimulus offset, subjects said a single noun describing what the hidden target was. We recruited 4 naive subjects (22 to 24 years old, 2 female), each one participating in 573 trials.

**SVM-category.**    To study the effect of object co-occurrences, we used a binary vector of size $1 \times C$ as input to a classifier, where the $i$th entry is 1 if there was an object from category $i$th in the image and 0 otherwise. We assume that the model had perfect information about *all* the object labels (all objects were visible except for the hidden target). A multi-class support vector machine (SVM) classifier was used to predict the hidden target based on this vector.

**SVM-category-instances.**    Extending SVM-category, we constructed a vector of size $1 \times C$ where the $i$th entry contained the number $n$ of instances of the $i$th category in the image. A multi-class SVM classifier was used to predict the hidden target based on this vector.

**Hidden Markov Model (HMM).**    To study the temporal dynamics over multiple clicks, we considered a Hidden Markov Model where we used all the training images in MSCOCO dataset to calculate the co-occurence matrix of size $C \times C$ as the transition probability matrix. We use

normalized uniform vector of size $1 \times C$ as the initial probability. We fine-tuned VGG16 on the MSCOCO dataset and used it for classifying the cropped region at the human clicked locations where the classification vector contributes to emission matrix. The Viterbi decoding algorithm Blunsom (2004) was used for making inferences about the hidden target.

**DeepLab-Conditional Random Field (CRF)** One interesting solution to reason about the hidden target is to run state-of-the-art semantic segmentation algorithms and use majority voting on the predicted labels over all pixels in the bounding box. We used the instantiation in DeepLab-CRF (Chen et al., 2017).

## C Upper bound: performance including object information

Our paper focuses on contextual reasoning with the target object completely occluded. It is interesting to provide an upper bound for performance where the object regions are revealed to ClickNet using the same images (**ClickNet-ObjRevealed**. The setup is as in the lift-the-flap problem but the initial condition is different: **ClickNet-ObjRevealed** is first presented with the tightest bounding box of the target object with the target revealed in high resolution and context blurred, and then the model has to click on the image in a sequential manner and all the previous clicked regions are deblurred. We fine-tuned the original ClickNet on this task and tested it on the same set of images but with the target objects revealed. The results are reported in Figure S4. With 1 click, **ClickNet-ObjRevealed** showed an accuracy of $66.1 \pm 0.88$ %. As more context is revealed (8 clicks), the accuracy of **ClickNet-ObjRevealed** increased to $71.7 \pm 0.84$ %.

In addition, we include another baseline **ClickNet-ObjOnly** where *only* the object is presented to **ClickNet-ObjOnly** in high resolution from the beginning and there is no context at all. In this case, the clicks have no relevance. We report the performance in Figure S4. With one click, **ClickNet-ObjRevealed** showed an accuracy of $47.3 \pm 0.93$ % and with 8 clicks the accuracy was $47.2 \pm 0.93$ %

As an additional upper bound comparison point we considered state-of-the-art object detection algorithms. In the literature, there are many works relying on context for object detection Chen et al. (2018b) and Redmon & Farhadi (2018). Here, we tested YOLO3 Redmon & Farhadi (2018) on these two cases (object revealed and object only). Since there are no recurrent connections in YOLO3, the reported results below assume YOLO3 can see full-resolution, i.e. no-blurring. In the object-revealed case, the accuracy was $65 \pm 2$ % and in the object-only case, the accuracy was $44 \pm 2$%.

There are several interesting observations from these upper bound measurements.

(i) As expected, revealing the object leads to better performance than the lift-the-flap condition with a hidden object. Yet, remarkably, contextual information above was essentially equivalent to object information alone (compare Figure 3 with the numbers above). Furthermore, revealing the object only increased performance by about 10-15 % with respect to the context only condition.

(ii) The accuracy of ClickNet-ObjectRevealed increases over clicks, which further confirms the important role of context in object recognition even when the object is shown.

(iii) ClickNet-ObjectRevealed and ClickNet-ObjectOnly slightly outperform YOLO3 in the object-revealed and object-only cases.

(iv) The summation of accuracies of ClickNet-ObjectOnly and ClickNet is not equal to ClickNet-ObjectRevealed which shows that the combination of context and object recognition is not linear.

(v) ClickNet-ObjectOnly is robust to time changes: although more clicks do not reveal any more information in this case, the recurrent connections did a good job in maintaining its accuracy (instead of forgetting what ClickNet has seen at the zero click).

## D Incongruent context

The observation that contextual information can help infer what a completely occluded object suggests that placing objects in a "wrong" context could impair recognition. Indeed, several

behavioral and computational tests have shown that objects are harder to recognize when they are out of context (Beery et al. (2018); Dvornik et al. (2018); Choi et al. (2012)). We verified these previous observations in a separate experiments where we constructed images where the objects were placed in incongruent contexts. To be consistent with the lift-the-flap case, we used the same test set from MSCOCO, cropped the objects and pasted them in either a congruent context (Figure S7(a)) or in an incongruent context (Figure S7(b)), as done in previous studies Choi et al. (2012). We tested these congruent and incongruent images both on ClickNet and humans. As expected, and consistent with previous work, we observed that performance in congruent images was higher than performance in incongruent images both for ClickNet (13.8% difference) and humans (18.4% difference).

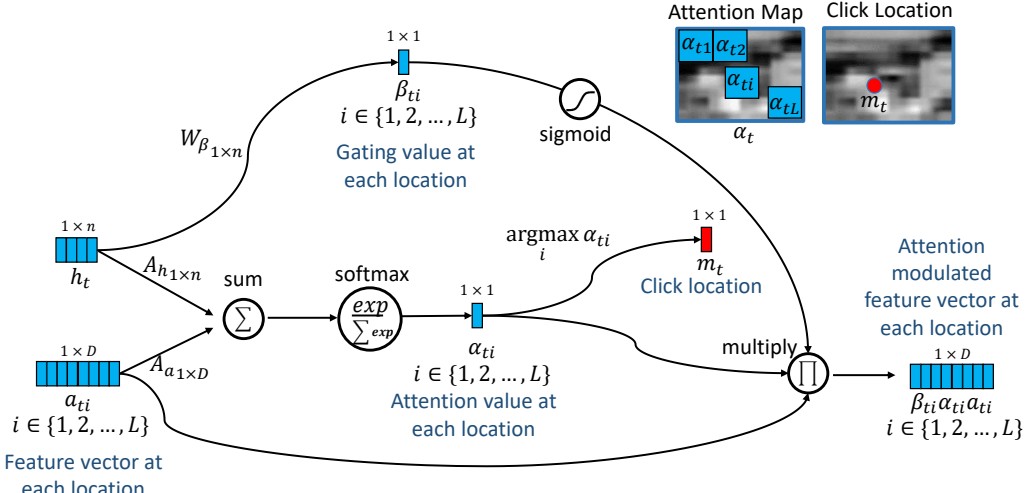

Figure S1: **Schematic illustration of the attention module implementation**. Expanding on the overall ClickNet architecture shown in Fig 2, here we zoom into the attention module. The attention module takes as inputs the features at each location $a_{ti}$ and the output of the LSTM module $h_t$ and selects the next click location $m_t$ and a map that modulates the features at each location (see Section 4 for a description of all the variables).

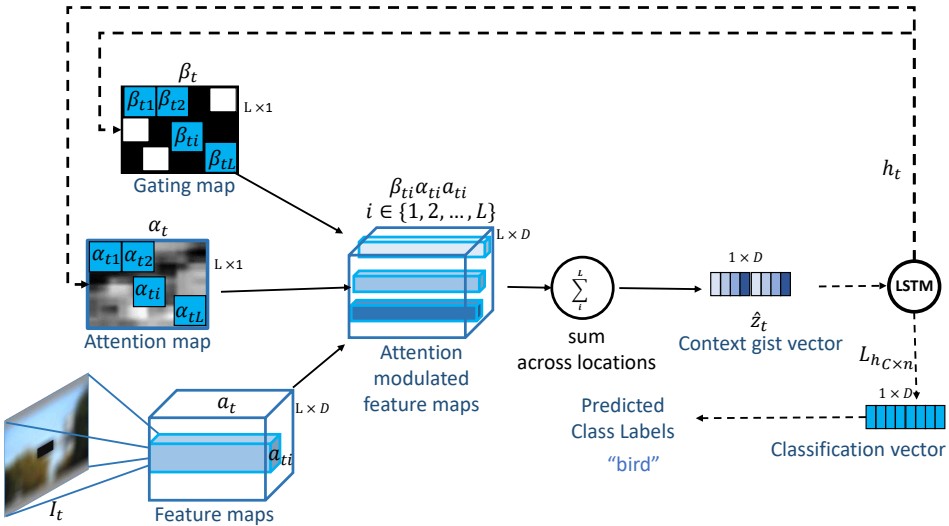

Figure S2: **Schematic illustration of the LSTM module implementation**. Expanding on the overall ClickNet archietcture shown in Fig 2, here we zoom into the LSTM module. The LSTM module takes as input context gist vector $\hat{\mathbf{z}}_{\mathbf{t}}$ and integrates the information with the previous state to inform the attention module in the next time step via $h_t$ and to predict a class label (see Section 4 for a description of all the variables).

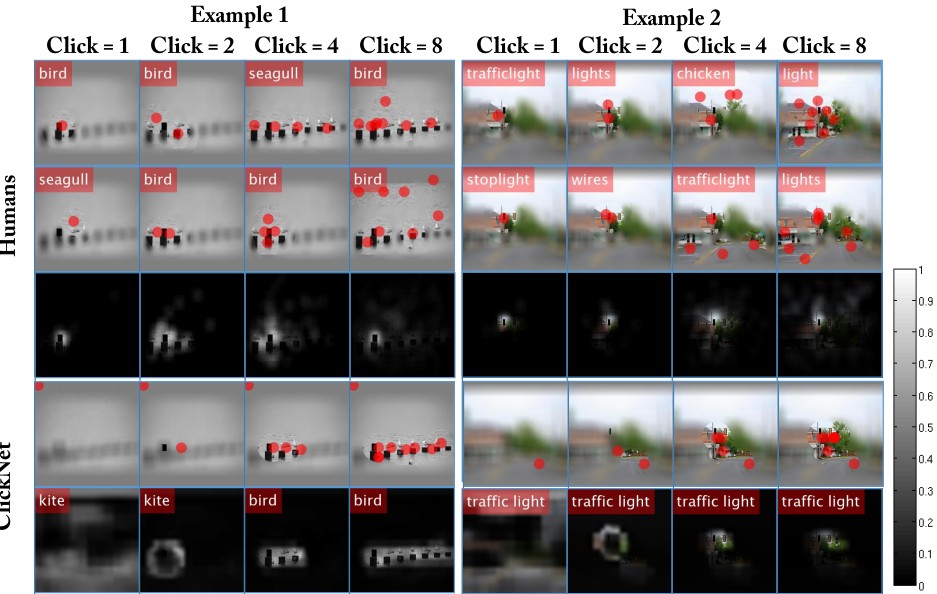

Figure S3: **Example visualziation for humans and ClickNet.** This figure shows click locations and attention maps using the same format as Fig 4, here adding results for 2 clicks and 4 clicks.

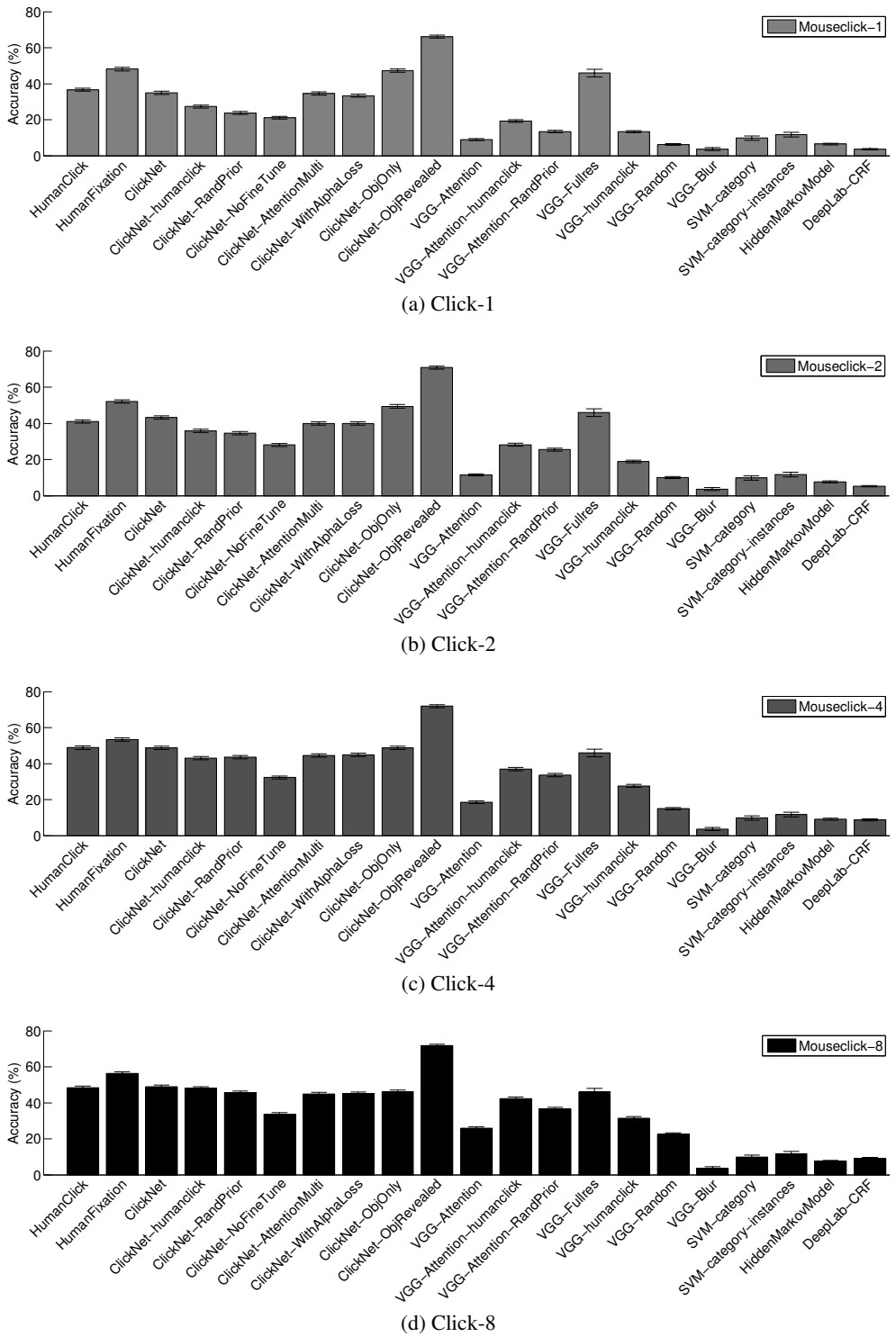

Figure S4: **Contextual reasoning accuracy of humans and models**. Expanding on the results in Fig. 3a-b, here we add the results for 2 clicks and 4 clicks, as well as additional comparative models (Section 4.5, Section A and Section B Section D) describe each model).

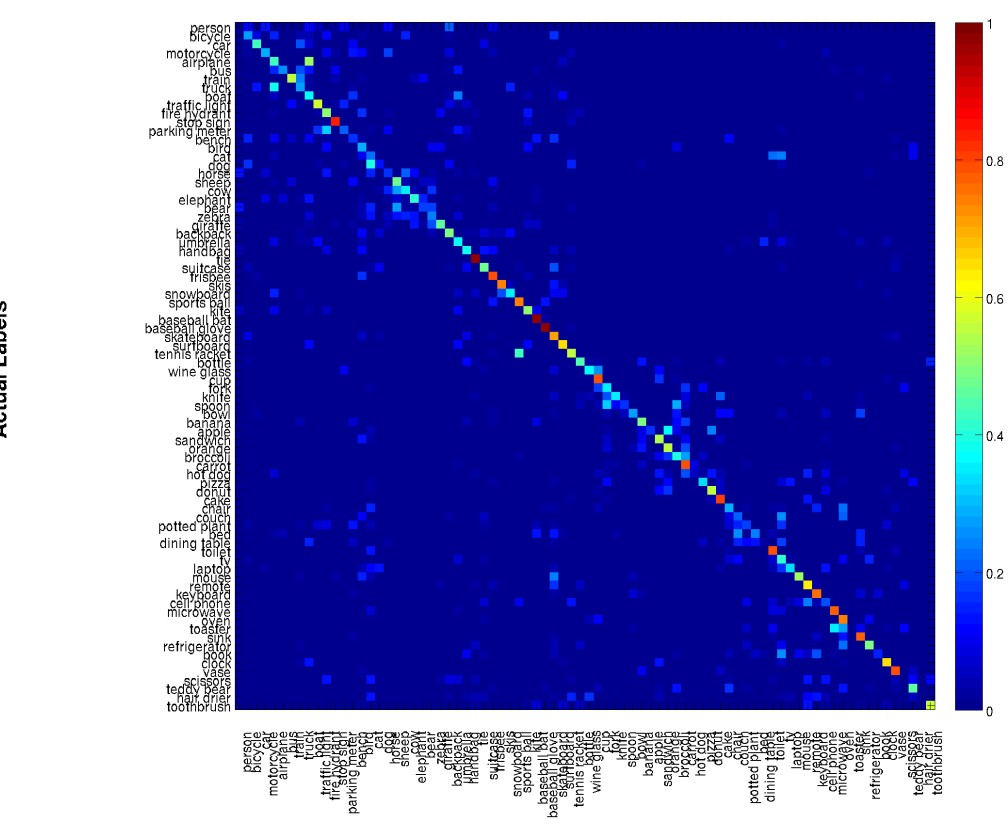

Figure S5: **Confusion matrix for ClickNet with all click conditions**. The format is the same as in Figure 5a, except showing all 80 categories here. The element in row $i$, column $j$ denotes the probability that ClickNet predicted label $j$ while the ground truth label was $i$ (see scale bar on right). The sum of all probabilities in a row equals 1.

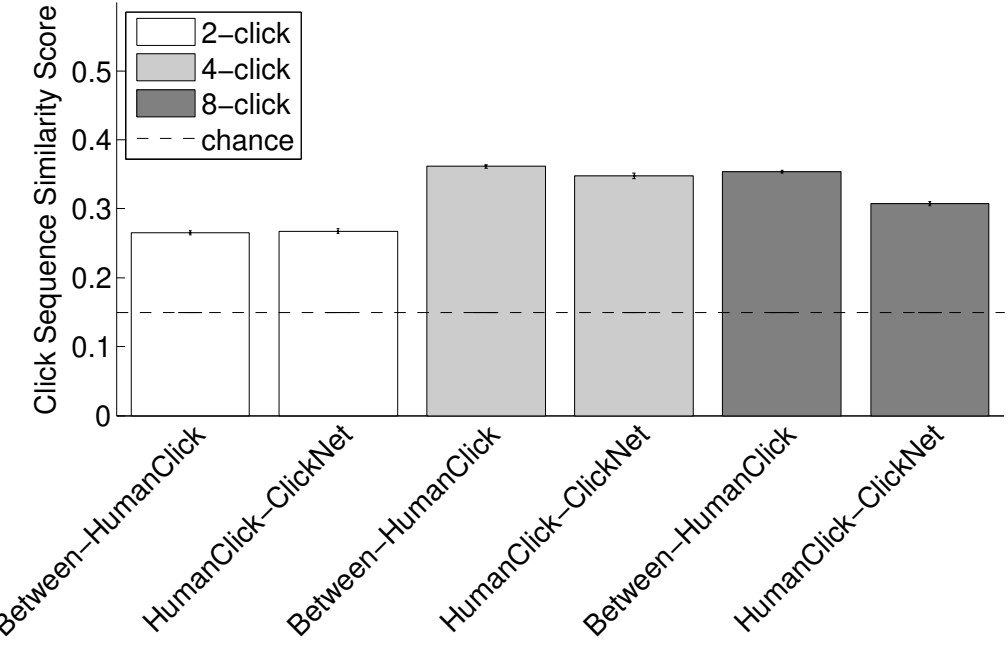

Figure S6: **Image-by-image consistency in the spatiotemporal pattern of click sequences across different number of clicks**. CLick sequence score is originally defined in evaluating eye fixation sequence consistency Madsen et al. (2012); Zhang et al. (2018). Here, we use the same metrics to compare the click sequences between-subjects and between ClickNet and subjects for 2 (light gray), 4 (dark gray), and 8 (black) clicks. The larger the click sequence score, the more similar the sequences are. The dashed line indicates chance performance, obtained by randomly permuting the clicks among images. Results shown here are averaged over subject pairs.

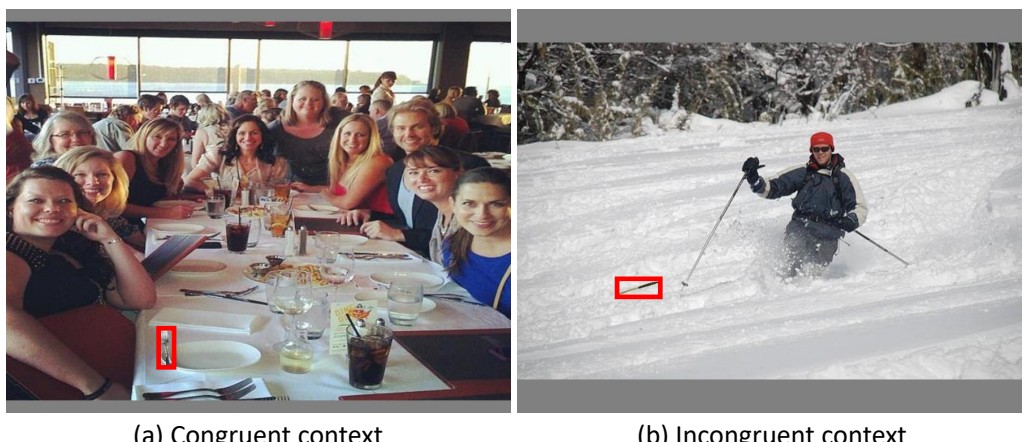

(a) Congruent context          (b) Incongruent context

Figure S7: **Example images from congruent and incongruent context**. (a) congruent example: a fork (from another image) pasted in a dining scene; (b) incongruent example: a knife (from another image) pasted in a ski field. Red bounding boxes indicate target object locations.

