# OpenReview forum: "Lift-the-flap: what, where and when for context reasoning"
_ICLR.cc/2020/Conference — Reject_

### Official Review · AnonReviewer2 · 2019-10-18
**Official Blind Review #2**

**Rating:** 3

**Review:**

The paper is about context reasoning in the visual recognition. They designed a task called 'lift-the-flap', where the human or the model are given an image with one of the object blacked out. The image is provided in a relative low resolution, and the subjects could choose to click on area to reveal a local window of high-res image. The subject then need to answer what is the object in the blacked out region after a few clicks (ranging from 1 to 8 clicks). The authors has collected human data using mTurk, and also trained a model (named ClickNet) to perform the task. ClickNet has a LSTM memory which can carry the information of previous clicks and use spatial attention mechanism to decide the next click location (just use the argmax of the attention weight.) I find the paper is clearly-written and quite easy to follow. I think this is an interesting paper comparing human attentional performance to a trained computation model (without any human-imposed prior), showing that they behave similarly. However, I am not sure it would be of general interest to the ICLR audience. This may be more suitable for cognitive science conference/journal in my opinion. I have a few suggestions below:

1. In formula 2, the attention weight (before softmax) is  " e_ti = A_h * h_{t−1} + A_a * a_ti " I am surprised there is no term that is h_{t-1} * a_ti, which do the content-based reading that is common in memory model like (DNC) or in language processing (transformer.) Can the author comment on this decision?

2. In result 5.1, the authors titled: "WHAT: IMPORTANCE SAMPLING VIA ATTENTION AND PRIOR INFORMATION". I am not sure I get where is the importance sampling appear in this paragraph. Importance sampling is a specific term, which refers to estimating properties of a particular distribution, while only having samples generated from a different distribution than the distribution of interest. I did not see how that is related to this paragraph. Or, I may have missed it, and would like the authors to elaborate.

3. I could not follow well the discussion in section 5.5 WHEN: ROLE OF RECURRENT CONNECTIONS. Based on the title of the section, I would expect to discuss why the recurrence of LSTM is important. Though, the paragraph is mainly talking about other control models like SVM and HMM, so I don't quite follow it. Also, it is not clear to me the word 'When' is a good choice. I am convinced that the LSTM helps the model carrying information across click, but it is not clearly shown the sequential order is critical (say click location 1, then location 2, then location 3 compared to location 2, location 3, then location 1 matters) Maybe it does? But, it is now shown in the results.

4. The comparison of where the human versus the model click (in Figure 6) is impressive. I wonder if the authors could go even one step further that is to see if the temporal order of the clicks are similar in human vs. model. It may not be the case. However, if it is true, that could make an even stronger case that the model has a similar way to decide attentional location compared to human subjects.

Minor:
1. It is impressive that the authors have considered a wide range of control models. Though, for a conference paper with limited page limit, I don't think it is necessary to describe all these models in the main text ( for the SVM, HMM, and CRF models.)


**Experience Assessment:**

I have read many papers in this area.

**Review Assessment: Checking Correctness Of Derivations And Theory:**

N/A

**Review Assessment: Checking Correctness Of Experiments:**

I carefully checked the experiments.

**Review Assessment: Thoroughness In Paper Reading:**

I read the paper at least twice and used my best judgement in assessing the paper.

---

> ### Author Response · Authors · 2019-11-11
> **Answers to R2 - Part 1**
>
> We thank the reviewer for constructive feedbacks. Answers are enclosed in sqared brackets. Page and figure numbers refer to the revised version of the paper.
>
> [R2]
>
> Review: The paper is about context reasoning in the visual recognition. They designed a task called ’lift-the-flap’, where the human or the model are given an image with one of the object blacked out. The image is provided in a relative low resolution, and the subjects could choose to click on area to reveal a local window of high-res image. The subject then need to answer what is the object in the blacked out region after a few clicks (ranging from 1 to 8 clicks). The authors has collected human data using mTurk, and also trained a model (named ClickNet) to perform the task. ClickNet has a LSTM memory which can carry the information of previous clicks and use spatial attention
> mechanism to decide the next click location (just use the argmax of the attention weight.) I find the paper is clearly-written and quite easy to follow. I think this is an interesting paper comparing human attentional performance to a trained computation model (without any human-imposed prior), showing that they behave similarly.
>
> [R2.1]
> However, I am not sure it would be of general interest to the ICLR audience. This may be more suitable for cognitive science conference/journal in my opinion.
>
> [Answer: As the reviewer points out, our work combines Cognitive Science and Computer Science. Our work directly concerns the type of computer vision experiments that have been critical to the development of recent AI approaches in pattern recognition and that computer vision has historically drawn on inspiration from Cognitive Science and Neuroscience. We hope that this work will strengthen these connections and bring insights to the computer vision community and more generally to the deep learning community. Several amazing works connecting cognitive neuroscience and computer science have been previously published in ICLR (e.g., Linsley et
> al, Learning what and where to attend,ICLR, 2019; Zheng et al, Revealing interpretable object representaitons from human behavior, ICLR, 2019).]
>
> [R2.2]
> In formula 2, the attention weight (before softmax) is eti= Ah × ht..1 + Aa × ati, I am surprised there is no term that is ht_1 × ati, which do the content-based reading that is common in memory model like (DNC) or in language processing (transformer.) Can the author comment on this decision?
>
> [Answer: In DNC, attention denotes the similarity between the writing or reading content with the content stored in each memory location. However, in ClickNet, it is not clear how each element in ht corresponds to the content in each channel of feature vector ati. We share the reviewer’s intuition that ht..1 and ati should be combined, but whether this should be an additive term or a multiplicative term is not obvious. To assess this question, we implemented an alternative model: as suggested by the reviewer, instead of addition, we introduced ht_1 × ati in equation 2 and reported the result in Figure S4, and discussed the results on page 5 in the main text and Section A in Supplemenatry Material. Overall, adding the multiplicative term did not improve the model’s performance.]
>
> [R2.2]
> In result 5.1, the authors titled: ”WHAT: IMPORTANCE SAMPLING VIA ATTENTION AND PRIOR INFORMATION”. I am not sure I get where is the importance sampling appear in this paragraph. Importance sampling is a specific term, which refers to estimating properties of a particular distribution, while only having samples generated from a different distribution than the distribution of interest. I did not see how that is related to this paragraph. Or, I may have missed it, and would like the authors to elaborate.
>
> [Answer: We thank the reviewer for clarifying the usage of importance sampling in statistics. The reviewer is correct in that we were not using this term correctly. What we intended to convey here is that subjects (and the model) sample (click) the image (i.e. choose specific locations) according to an internal model of what is important or not. To avoid confusion, we have rephrased the title as ”WHAT: REGION SELECTION VIA ATTENTION AND PRIOR INFORMATION” in the revised version.]

---

> ### Author Response · Authors · 2019-11-11
> **Answer to R2 - Part 2**
>
> [R2.3]
> I could not follow well the discussion in section 5.5 WHEN: ROLE OF RECURRENT CONNECTIONS. Based on the title of the section, I would expect to discuss why the recurrence of LSTM is important. Though, the paragraph is mainly talking about other control models like SVM and HMM, so I don’t quite follow it. Also, it is not clear to me the word ’When’ is a good choice. I am convinced that the LSTM helps the model carrying information across click, but it is not clearly shown the sequential order is critical (say click location 1, then location 2, then location 3 compared
> to location 2, location 3, then location 1 matters) Maybe it does? But, it is now shown in the results.
>
> [Answer: The reviewer is correct. We did not test whether the order matters or not. We asked subjects (and the model) to perform a fixed number of clicks and then report the inferred hidden object. Thus, we cannot distinguish click location 1 (CL1) followed by CL2 from CL2 followed by CL1 in terms of recognition accuracy. We note that we are not making any claims about the relevance of the click order. What we can do is compare the sequence of clicks between different subjects, and between humans and models. In response to R2.4 below, we have now carried out additional analyses to quantify this question (see below). Our nomenclature in this section was unclear. What
> we mean by “when” is that the temporal integration of information provided by LSTM is important. We have now clarified this in Section 5.5 and Section 6 on page 10. This section was intended to provide evidence that alterntive models that lack this temporal integration were not as effective]
>
> [R2.4]
> The comparison of where the human versus the model click (in Figure 6) is impressive. I wonder if the authors could go even one step further that is to see if the temporal order of the clicks are similar in human vs. model. It may not be the case. However, if it is true, that could make an even stronger case that the model has a similar way to decide attentional location compared to human subjects.
>
> [Answer: This is also a very interesting question. As suggested by the reviewer, we compared the spatiotemporal dynamics of click sequences between human and ClickNet using a click sequence score. This score was inspired by metrics used to compare DNA sequences and has been used to evaluate the similarity of two eye fixation sequences. (Madsen, Adrian, etal Using ScanMatch scores to understand differences in eye movements between correct and incorrect solvers on physics problems, 2012, ACM; Zhang et al Nature Communications 2018). We reported the results in Figure S6 and we discuss the results on page 4 and Section 5.3. Overall, ClickNet was consistent with the spatiotemporal sequence of human fixations, significantly more so than expected by chance.]
>
> [R2.5]Minor:1. It is impressive that the authors have considered a wide range of control models. Though, for a conference paper with limited page limit, I don’t think it is necessary to describe all these models in the main text ( for the SVM, HMM, and CRF models.)
>
> [Answer: We have moved these models to the supplementary material in the revised version, as
> suggested.]

---

> > ### Comment · AnonReviewer2 · 2019-11-15
> > **Thank you for a detailed reply to the comments.**
> >
> > Thank you for a detailed reply to the comments. I think the manuscript would be improved after the suggested changes. I still find the results too be a bit on the weak (not so exciting) side. Though, I appreciate the work is quite solid and well-written. I think it is on the borderline of acceptance.

---

> > > ### Author Response · Authors · 2019-11-15
> > > **Thank you**
> > >
> > > Thank you very much for your feedback and comments. We appreciate you taking the time to go through the text and responses. Please let us know if there are any other technical questions or concerns that we can clarify.

---

### Official Review · AnonReviewer3 · 2019-10-21
**Official Blind Review #3**

**Rating:** 3

**Review:**

I agree that context is important in some visual recognition tasks. I think this is an ambitious study and find the employed experimental methods interesting.

However, it is not clear to me what has been actually revealed by this study.

In the last section, there is a statement “the model adequately predicts human sampling behavior and reaches almost human-level performance in this contextual reasoning task.” I think that at least the first half is overstatement; it is not well validated by the experimental results.

For instance, contrary to the authors’ claim, I do not think the effectiveness of the recurrent connections in ClickNet is sufficiently validated. I think a yet another baseline is missing in the experiments, which is a strategy of clicking points in the periphery of the black box while avoiding (or minimizing) overlaps of the regions deblurred by previous clicks.

Although ClickNet-RandPrior appears to be close to this strategy, it does not seem to use any constraint of avoiding such region overlap. On the other hand, in the training of ClickNet, a constraint \sum_t\alpha_{ti}=1 is used, which seems to play this very role, i.e., making it possible for ClickNet to avoid the region overlap. Isn’t the good performance of ClickNet fully attributable to this constraint?

The proposed method is to make subjects (or ClickNet) click a series of points in the input image and then deblur the local regions around the points. I suppose this procedure is accumulative, that is, once a local region is deblurred, it will be kept deblurred in the subsequent clicks. Then, I'm not sure if the order of clicking points really matters, whether it is a human subject or the ClickNet. For instance, is there any evidence that a click is dependent on the previous clicks, other than the above behavior of avoiding overlaps?

Additionally, I am somewhat skeptical if a pretrained VGG can really extract useful features from (partially) blurred images, even though it is not trained on blurred images. It is widely known that CNNs for visual recognition tasks are vulnerable to image blur, noise, etc. when they do not exist in the images used for training, which is a kind of domain shift.


**Experience Assessment:**

I have published in this field for several years.

**Review Assessment: Checking Correctness Of Derivations And Theory:**

N/A

**Review Assessment: Checking Correctness Of Experiments:**

I carefully checked the experiments.

**Review Assessment: Thoroughness In Paper Reading:**

I read the paper at least twice and used my best judgement in assessing the paper.

---

> ### Author Response · Authors · 2019-11-11
> **Answers to R3 - part 1**
>
> We thank the reviewer for constructive feedbacks. Answers are enclosed in squared brackets. Page and figure numbers refer to the revised version of the paper.
>
> [R3]
>
> Review: I agree that context is important in some visual recognition tasks. I think this is an ambitious study and find the employed experimental methods interesting. However, it is not clear to me what has been actually revealed by this study.
>
> [R3.1]
> In the last section, there is a statement “the model adequately predicts human sampling behavior and reaches almost human-level performance in this contextual reasoning task.” I think that at least the first half is overstatement; it is not well validated by the experimental results.
>
> [Answer: We have rewritten this statement to remove the ambiguous term ”adequately”. Now we write: ”... the model approximates human sampling behavior (Click consistency in spatial domains (Euclidean distance distribution and spatial bias in Figure 6) and temporal domains (click sequence score in Figure S6)) and reaches almost human-level performance in this contextual reasoning task (contextual reasoning accuracy in Figure 3 and Figure 5b).” We note that we are strictly referring to this particular task and we are not claiming that the model generally describes human sampling behavior in other domains. ]
>
> [R3.2]
> For instance, contrary to the authors’ claim, I do not think the effectiveness of the recurrent connections in ClickNet is sufficiently validated. I think a yet another baseline is missing in the experiments, which is a strategy of clicking points in the periphery of the black box while avoiding (or minimizing) overlaps of the regions deblurred by previous clicks. Although ClickNet-RandPrior appears to be close to this strategy, it does not seem to use any constraint of avoiding such region overlap.
>
> [Answer: We have removed the regularization constraint in Equation 9 (see R3.3). There is no difference between ClickNet and ClicNet-RandPrior in whether the clicks can potentially overlap. However, we note that the chance that two random clicks near the bounding box overlap is very small (p=0.0065 using the same overlap criteria defined for human clicks, page 6). We further note, that two clicks can also overlap in ClickNet.]

---

> ### Author Response · Authors · 2019-11-11
> **Answers to R3 - Part 2**
>
> [R3.3]
> On the other hand, in the training of ClickNet, a constraint alpha_ti = 1 is used, which seems
> to play this very role, i.e., making it possible for ClickNet to avoid the region overlap. Isn’t the good
> performance of ClickNet fully attributable to this constraint?
>
> [Answer: We especially thank the reviewer for making this point. As the reviewer suggested, we removed the regularization term implemented by the alpha loss term in Equation 9 (ClickNet-noalphaloss). In contrast with the original model proposed in an image captioning task (Xu Kevin et al, Show, attend and tell: Neural image caption generation with visual attention, 2015, ICML), the results of ClickNet-noalphaloss are actually better than those in the original ClickNet including the alpha loss constraint (Figure S4). Since ClickNet-noalphaloss is more efficient and
> more elegant than the original ClickNet, we revised Equation 9 in the main text, simplified the model and updated all the corresponding figures, and results in the revised paper. Overall, three pieces of evidence show that ClickNet learns more than avoiding click overlaps: (i) the increase in performance with alpha loss removed indirectly hints that the recurrent connections in the LSTM learn not only to avoid click overlaps but also to integrate information across multiple clicks (Figure S4); (ii) ClickNet-noalphaloss still outperforms ClickNet-RandPrior even when controlling for the spatial distribution of random click locations and even though the very small number of overlapping clicks is equivalent for these two models (Figure 3); (iii) We show that the sequence pattern of clicks
> made by ClickNet is consistent with human click sequence patterns (Figure S6).]
>
> [R3.4]The proposed method is to make subjects (or ClickNet) click a series of points in the input image and then deblur the local regions around the points. I suppose this procedure is accumulative, that is, once a local region is deblurred, it will be kept deblurred in the subsequent clicks. Then, I’m not sure if the order of clicking points really matters, whether it is a human subject or the ClickNet. For instance, is there any evidence that a click is dependent on the previous clicks, other than the above behavior of avoiding overlaps?
>
> [Answer: The reviewer correctly interpreted that once a region is deblurred, it stays deblurred. We note that we are not making any statement about the order in which the clicks occurred. Because of the experimental setup, we cannot really test whether the order of clicks matters for recognition accuracy. In any given trial, the number of clicks was fixed, and we do not probe recognition before the total number of clicks for that trial is executed. Thus, we do not really know whether click location 1 (CL1), followed by CL4, then CL3, then CL2 would lead to a different recognition accuracy than CL2, CL4, CL3, CL1. Inspired by this question, we compared the spatiotemporal dynamics of click sequences, using metrics similar to the ones used to compare DNA sequences, also used to evaluate the similarity of two eye fixation sequences (Madsen etal Using ScanMatch scores to understand differences in eye movements between correct and incorrect solvers on physics problems, 2012, ACM; Zhang et al, Nature Communications 2018). The spatiotemporal sequence of clicks was more consistent between subjects than expected by chance (Figure S6). In the above example, the degree of consistency in CL1, CL4, CL3, CL2 was higher than expected by chance. Furthermore, the consistency in the spatiotemporal sequence of clicks between ClickNet and humans was also higher than expected by chance (Figure S6). In sum, we do not know, and hence we do not
> claim, whether the click order impacts recognition accuracy, but we do know that click order is not random and the model captures this non-random behavior.]
>
> [R3.5]
> Additionally, I am somewhat skeptical if a pretrained VGG can really extract useful features from (partially) blurred images, even though it is not trained on blurred images. It is widely known that CNNs for visual recognition tasks are vulnerable to image blur, noise, etc. when they do not exist in the images used for training, which is a kind of domain shift.
>
> [Answer: The reviewer is right about the domain shift in the case of VGG16 pre-trained with full resolution images directly used for feature extraction. We included fine-tuning of VGG16 during training CLickNet as mentioned in the Implementation Details (page 6). To evaluate whether fine-tuning pre-trained VGG16 is necessary in the lift-the-flap task, we reported the results of an ablated model (ClickNet-nofinetuned) on Figure S4 (also see page 4 in the main text and Section A in the supplementary material), which was 14% lower than ClickNet. ]

---

### Official Review · AnonReviewer1 · 2019-10-30
**Official Blind Review #1**

**Rating:** 6

**Review:**

The paper does a psychological-computational-combined experiments for context reasoning. The experiment is done by "lift-the-flap" -- masking out a region of interest in the image and let either the human or a convNet based model to guess what it is by checking the context. Both of them are first shown with a blurred image with masked region, and then start to guess by clicking on surrounding regions and unblur them. A lot of baselines are compared and it is shown that the computational model is working well, and the behavior is highly correlated with humans.

+ The paper reads very well, the illustrations and tables are very well done.
+ The experiment itself is interesting that it delves into the context problem directly. It would be interesting to see how it works for objects that are out of context though:
http://people.csail.mit.edu/myungjin/outOfContext.html
+ A like it that a great set of baselines and analysis are done for the paper. It strengthens the paper a lot.

- I think the paper needs to have a baseline for the upper bound as well: what is the accuracy if the region is seen? In other words, what is the performance if no context is needed. It would be interesting to see the gap over there. A baseline could be a region-classification model, e.g., from:
Chen, Xinlei, et al. "Iterative visual reasoning beyond convolutions." Proceedings of the IEEE Conference on Computer Vision and Pattern Recognition. 2018.
Just from the results (e.g. Fig 3) it seems our current models are already doing pretty well! (and it is VGG, not the best model yet), but on the other hand it is not clear how such models can help recognize visible objects better -- maybe a lot of the things that the context can offer has already been incorporated in the object pixels itself.

**Experience Assessment:**

I have published one or two papers in this area.

**Review Assessment: Checking Correctness Of Derivations And Theory:**

N/A

**Review Assessment: Checking Correctness Of Experiments:**

I assessed the sensibility of the experiments.

**Review Assessment: Thoroughness In Paper Reading:**

I read the paper at least twice and used my best judgement in assessing the paper.

---

> ### Author Response · Authors · 2019-11-11
> **Answers to R1**
>
> We thank the reviewer for constructive feedbacks. Answers are enclosed in squared brackets. Page and figure numbers refer to the revised version of the paper.
>
> [R1]
>
> [R1.1]
> The experiment itself is interesting that it delves into the context problem directly. It would be interesting to see how it works for objects that are out of context though: http://people. csail.mit.edu/myungjin/outOfContext.html
>
>
> [Answer: This is a great idea. As suggested, in the follow-up work after ICLR submission, we conducted another human behavioral experiment to test recognition accuracy in congruent versus incongruent backgrounds, and we also evaluated ClickNet in this new task. We added this experiment in the supplementary material (Figure S7 and Section D). We also discuss this point in the main text (page 2) and cite this work (page 2 and 3) in the revised version of our paper.]
>
> [R1.2]
> I think the paper needs to have a baseline for the upper bound as well: what is the accuracy if the region is seen? In other words, what is the performance if no context is needed. It would be interesting to see the gap over there.
>
> [Answer: As the reviewer suggested, there are two interesting questions here: (1) Performance for the same image without the black box: We fine-tuned ClickNet on the same training images with the target object revealed ClickNet-ObjRevealed), shown in Figure S4 in the revised version. (2) Performance for the object only condition: We tested ClickNet fine-tuned above in (1) on the same test set but only showing the object without context and reported performance (ClickNet-ObjOnly) in Figure S4 in the revised version. Please see page 7 in the main text and Section C in the supplementary material in the revised paper for analyses and discussion.]
>
> [R1.3]
> A baseline could be a region-classification model, e.g., from: Chen, Xinlei, et al. ”Iterative visual reasoning beyond convolutions.” Proceedings of the IEEE Conference on Computer Vision and Pattern Recognition. 2018.
>
> [Answer: This is another great suggestion. Despite our attempts, we were not able to get the publicly available source code of this paper to run in the short time since this suggestion. Due to the limited time, we chose an alternative baseline, YOLO3 (Redmon J et al, YOLOv3: An Incremental Improvement), which is a state-of-the-art object detection algorithm. We are aware that this baseline is not comparable with the nice work of Chen et al, where there is a local spatial memory module to update belief and a global graph reasoning module, but we hope YOLO3 provides a general comparison to how ClickNet-ObjOnly and ClickNet-ObjRevealed perform on this task. YOLO3 was tested on full resolution test images with either objects revealed or object-only conditions (R1.2)
> yielding 65% and 44.2% performance, respectively. (Secition C). Meanwhile, we cited Chen’s
> work in the related work section, and we will continue to work on its implementation; we agree that
> this is an interesting comparison.]
>
> [R1.4]
> Just from the results (e.g. Fig 3) it seems our current models are already doing pretty well! (and it is VGG, not the best model yet), but on the other hand it is not clear how such models can help recognize visible objects better – maybe a lot of the things that the context can offer has already been incorporated in the object pixels itself.
>
> [Answer: We agree with the reviewer that in object recognition tasks in computer vision, VGG16 and other feed-forward networks implicitly but heavily rely on contextual information to recognize objects (e.g., Linsley et al, LEARNING WHAT AND WHERE TO ATTEND,ICLR, 2019). Some studies (Beery et al, Recognition in terra incognita, 2018, ECCV; Dvornik, Nikita, Modeling visual context is key to augmenting object detection datasets, 2018, ECCV) and our work following up on the suggestion in R1.1 also show that these models can fail miserably when objects are placed in an incongruent context. These studies motivate us to systematically examine the role of context in
> lift-the-flap tasks. Furthermore, in situations of complete occlusion (lift-the-flap), here we provide human and computational benchmarks for object inference. We have added these references and points in the text (page 2)].

---

### Author Response · Authors · 2019-11-11
**Paper has been revised. Answers to the reviews have been posted**

We thank the reviewers for constructive feedbacks. Answers are enclosed in squared brackets. Page and figure numbers
refer to the revised version of the paper.

All revisions in the revised paper have been highlighted in orange for easy reference.

---

### Decision · Program_Chairs · 2019-12-19

**Decision:**

Reject

**Comment:**

The authors present the task lift-the-flap where an agent (artificial or human) is presented with a blurred image and a hidden item. The agent can de-blur the parts of the image by clicking on it. The authors introduce a model for this task (ClickNet) and they compare this against others.
As reviewers point, this paper presents an interesting set of experiments and analyses. Overall, this type of work can be quite influential as it gives an alternative way to improve our models by unveiling human strategies and using those as inductive biases for our models. That being said, I find the conclusions of this paper quite narrow for the general audience of ICLR (as R2 and R3 also point), as authors look into an artificial task and show ClickNet performs well. But what have we learned beyond that? How do we use these results to improve either our models or our understanding of these models? I believe these are the type of questions that  are missing from the current version of the paper and that if answered would greatly increase its impact and relevance to the ICLR community. At the moment though, I cannot recommend this paper for acceptance.